# Neighbourhood environments for a healthy lifestyle among young single-person households experiencing housing poverty in Seoul, South Korea: a spatiotemporal qualitative study protocol

Dong Ha Kim [ID],[1] Jihyun Lee [ID],[2] Seunghyun Yoo [ID] [1,2]

¹Institute of Health and Environment, Seoul National University, Gwanak-gu, Seoul, South Korea
²Department of Public Health Sciences, Graduate School of Public Health, Seoul National University, Gwanak-gu, Seoul, South Korea

**Correspondence to**
Dr Seunghyun Yoo;
syoo@snu.ac.kr

## ABSTRACT

**Introduction** The number of single-person households is increasing globally—including in South Korea, where they account for over 30% of all households. Young single-person households in South Korea face health problems and housing challenges. Both the perceived and objective aspects of the neighbourhood environment, as a community asset, play a significant role in sustaining a healthy lifestyle. This study aims to explore and describe the meaning, roles and spatiotemporal characteristics of neighbourhood environments for a healthy lifestyle in young single-person households experiencing housing poverty in Seoul, South Korea.

**Methods and analysis** This ongoing study uses an extended qualitative geographic information systems approach to explore a district in the city of Seoul that has the highest population density of young single-person households experiencing housing poverty. The study sample comprises young single-person households aged 19–39 years who are experiencing housing poverty in the study area, with an expected saturation point of approximately 55 participants. We employ online and offline recruitment strategies to ensure the inclusion of diverse perspectives and a multimethod approach that combines descriptive and spatiotemporal data collection techniques (eg, individual in-depth interviews, field observations and mobile global positioning system tracking). The data analysis encompasses thematic and content analyses to understand the neighbourhood environment's perceived attributes and the spatiotemporal characteristics of healthy lifestyles. In the integrated analysis, we plan to combine the qualitative findings with living space and daily-life patterns using qualitative software and a hybrid relational database.

**Ethics and dissemination** The Institutional Review Board of Seoul National University approved the research protocol on 18 May 2021. The findings will be shared at international conferences and published in academic journals. Additionally, an online seminar will be conducted to share the results with policy-makers, researchers, community organisations and health workers working with young single-person households experiencing housing poverty.

## STRENGTHS AND LIMITATIONS OF THIS STUDY

⇒ This study provides valuable insights by exploring the lived experiences of young single-person households experiencing housing poverty.

⇒ This study helps to identify the meanings and values attached to young single-person households experiencing housing poverty in various places, and to understand how these meanings are formed and interpreted.

⇒ This study may be time-consuming, labor-intensive and costly due to the need to collect and analyse data from multiple sources and devices.

⇒ This study requires a high level of competence and expertise in managing data quality to address the complexities of integrating diverse data sources and to ensure the active participation of the target population.

⇒ The interpretation of the study results should consider the characteristics and context of both the study area and participants, particularly in relation to housing poverty.

## INTRODUCTION

Single-person households are becoming increasingly common worldwide. In European Union member states, the proportion of single-person households increased from 31% in 2010 to 34% in 2017.[1] According to the United States Census Bureau,[2] the proportion of single-person households in the USA increased from 16.7% in 1969 to 28.4% in 2019, and the rate of increase is expected to accelerate over the next decade. Similar to most developed countries, South Korea (hereafter, Korea) has experienced notable household fragmentation in recent years, leading to a trend towards individualised life courses. In 2000, single-person households accounted for only 15.5% of all households in

Korea, but had the highest household composition ratio (27.2%) and overtook multiperson households in 2015. This figure had risen even further to 33.4% by 2021.[3]

Compared with the other demographic groups in Korea, young single-person households face a higher risk of health and housing in 2020, 63.2% of young single-person households aged 19–29 were either obese or overweight, 27.7% showed symptoms of depression and 29.3% were smokers.[4] These figures were statistically significantly higher than those for multiperson households.[5] In 2021, the Korea Research Institute for Human Settlements[6] reported that 60% of young single-person households spent more than 30% of their monthly income on housing rent, which is five times higher than that of young couple households and households with young adults living with parents. Additionally, according to the 2020 Korea Housing Survey,[7] 9.6% of single-person households living in houses with less than 14 m$^2$ of residential space experienced difficulties in soundproofing, ventilation, heating and kitchen facilities. These housing conditions are lacking compared with the international minimum housing standards that provide specific facilities and layouts such as kitchens, bathrooms, privacy protections, appropriate hot water supply temperatures and 25 m$^2$ of space per person.[8] This suggests that young single-person households in Korea may not have sufficient resources within their homes to manage and protect their health adequately. Given the challenges faced by these households, namely, insufficient home resources to adequately address health needs, highlighting the health-promoting factors of the neighbourhood environment could be a pivotal strategy for improving the quality of life of those living in single-person households.

Recognising the unique challenges faced by young, single-person households underscores the urgency of prioritising and cultivating a healthy lifestyle. A healthy lifestyle, intricately intertwined with the surrounding environment, is a holistic concept that transcends physical health, encompassing mental health, nutrition and overall quality of life.[9] The combination of factors such as regular exercise, a balanced diet, mental well-being and positive social interactions not only contributes to individual well-being but also interfaces with the neighbourhood environment.[9] This symbiotic relationship operates synergistically within the holistic framework of a healthy lifestyle. Recognising the interconnectedness of these elements is critical, as together they form the foundation for wellness and fortification against potential health risks, with the neighbourhood environment playing a pivotal role in shaping and supporting these health-promoting behaviours.[10]

*Neighbourhood environment* refers to the physical, social, economic and cultural factors that surround and influence residents within a specific geographical area.[11] These factors include land use patterns, street safety, public transportation systems, living facilities, parks and green spaces, housing quality, and social and cultural opportunities. Additionally, the neighbourhood environment has perceived and objective aspects.[12] The *perceived aspect* of a neighbourhood environment refers to how individuals perceive and experience their neighbourhood.[13] In contrast, the *objective aspect* of the neighbourhood environment refers to the physical and structural characteristics of the neighbourhood that can be measured quantitatively.[13] The perceived and objective aspects of the neighbourhood environment are significant in shaping individuals' experiences and health outcomes. Perceived aspects influence how individuals behave and engage with their communities, whereas objective aspects affect access to resources and services, safety and overall well-being.[14] By understanding the perceived and objective aspects of the neighbourhood environment, policy-makers, researchers and community members can collaborate to create healthier and more livable neighbourhoods.

Collecting data on various aspects, such as time, space and experience, is crucial for comprehensively understanding the perceived and objective neighbourhood environments related to healthy lifestyles. The time available for using the neighbourhood environment can shape patterns of participation in health behaviours,[15] and spatial access to neighbourhood environments can affect an individual's ability to maintain a healthy lifestyle.[16] Perceptions of neighbourhood environments can also affect intentions, practices and attitudes towards health behaviours.[17] Therefore, considering these aspects in data collection can provide a more comprehensive understanding of the relationship between the neighbourhood environment and healthy lifestyle of young single-person households experiencing housing poverty.

This study aims to explore and describe the meaning, roles and spatiotemporal characteristics of neighbourhood environments in order to secure healthy lifestyles for young single-person households experiencing housing poverty in Seoul, South Korea. The specific research questions are as follows:

1. What perspectives do young single-person households experiencing housing poverty have on their homes and neighbourhoods in terms of what would promote a healthy lifestyle?
2. How do young single-person households experiencing housing poverty maintain or attempt to pursue a healthy lifestyle in their neighbourhoods?
3. What positive attributes and characteristics of neighbourhood environments are necessary to support healthy lifestyles in young single-person households experiencing housing poverty?
4. What is the context and scope of perceived neighbourhood boundaries for young single-person households experiencing housing poverty?

## METHODS AND ANALYSIS
### Study design
This study employs an extended qualitative GIS (QGIS) research design by adding time-use data from daily life. The original QGIS methodology integrates qualitative

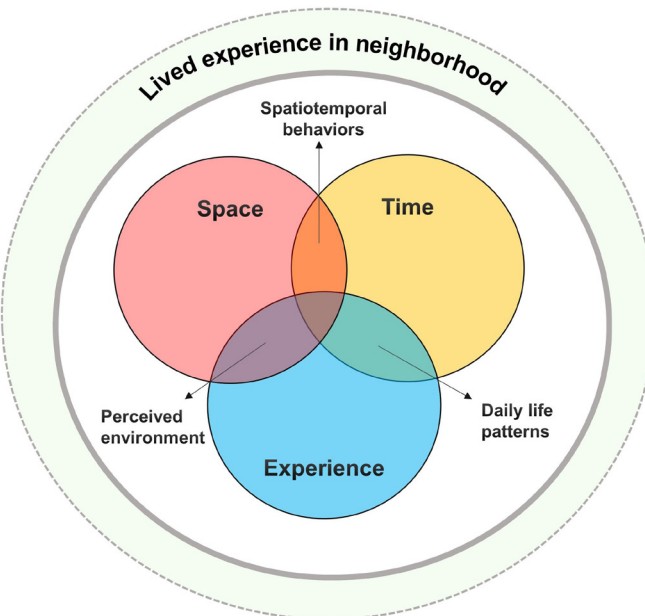

**Figure 1** Schematic diagram of this study.

and geospatial methods to investigate the rich contextual details of spatial experiences in various formats.[18] By merging qualitative methods with GIS-based techniques, QGIS provides an analytical framework for narrative inquiries into spatial experiences that have previously been unfeasible.[19 20] One advantage of visualising textual information on a map (eg, perceptions, attitudes, emotions, beliefs and behaviours) is the ability to explore spatial and environmental contexts that may not be fully elucidated

through interviews alone.[21] By incorporating time aspects into these benefits, we expect to explore the daily time usage patterns of young single-person households experiencing housing poverty in their neighbourhoods.

Figure 1 shows a schematic diagram of the extended QGIS used in this study. This diagram demonstrates how qualitative, spatial and temporal data can be combined through simultaneous processing to produce a rich contextual understanding of people's lived experiences in their neighbourhoods. This distinctive aspect sets our study's strategy apart from a mixed methods approach[22] as it has a unique sequence and arrangement for collecting both qualitative and quantitative data. Additionally, we take a more qualitative-oriented approach compared with traditional QGIS by interpreting GIS information based on patterns rather than concentrating solely on size, frequency and density.

## Setting

Our study is currently being conducted in an administrative district with the highest density of young single-person households experiencing housing poverty, Gwanak-gu, which is located in southern Seoul (figure 2). The district has three administrative neighbourhoods with a total area of 29.6 km$^2$, 60% of which is occupied by mountainous terrain.[23] As of September 2021, there were 164000 single-person households in the district, making it the region with the largest number of single-person households in Korea.[23] The main housing type in the district is the multiple complex house with a semi-basement, accounting for 55% of the total houses.[23] The

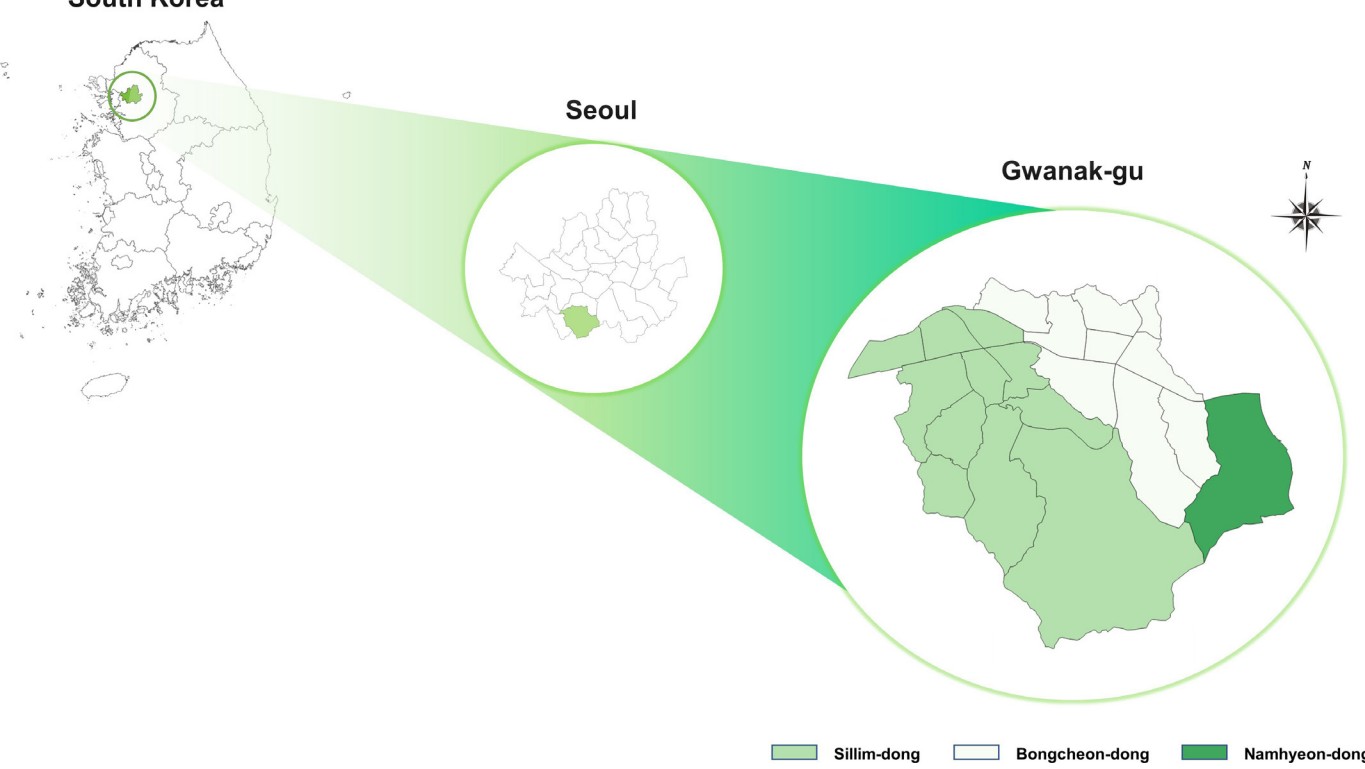

**Figure 2** Study site.

average per capita housing area was 28.2 m² (approximately 303 ft²), and the monthly rent for a studio apartment was about 400 000 won (approximately US$299.00) in 2021. Gwanak-gu is known for its poor and low-cost residential environment.[23] In addition, the district has a low-income community ('Go-si-chon' in Korean) where a large number of unemployed young people who prepare for the civil service examinations live in close quarters.[23]

## Study sample and recruitment

Three administrative neighbourhoods in Gwanak-gu, where young single-person households are concentrated, are included in our research as study areas. We are currently in the planning phase of conducting stratified purposive sampling[24] for single-person households (aged 19–39) experiencing housing poverty within the study areas. In this study, housing poverty is defined as residing in housing that either (1) fails to meet the minimum housing standards in Korea, or (2) requires more than 30% of and individual's monthly household income.[25] The exclusion criteria are individuals who have lived in the study areas for less than 6 months and those who do not possess electronic and digital devices capable of conducting online interviews. Online and offline recruitment strategies are currently being employed to ensure the inclusion of diverse perspectives from participants of various genders, ages, employment statuses, housing types and residence periods. Online recruitment involves posting recruitment documents on social network service platforms commonly used by young people in the study areas with the cooperation of online community managers. Offline recruitment entails attaching recruitment documents to telephone poles,

bulletin boards and walls located in areas frequented by local young people, as well as visiting public service centres that rent spaces to local young people. Participants will be recruited until data saturation is achieved; that is, when new information no longer produces significant changes to the code book.[26] The saturation point of study participants is expected to be approximately 9–17 people per neighbourhood, based on previous research,[27] and with additional participants to confirm saturation, the number of study participants is expected to be approximately 55.

## Data collection

We are currently employing a multimethod approach to gather descriptive and spatiotemporal information on the neighbourhood environment to determine what would be conducive for healthy lifestyles among young single-person households experiencing housing poverty. This approach combines descriptive data collection techniques, such as individual in-depth interviews, spatial data collection techniques (eg, cognitive mapping, field observations and mobile GPS tracking), and temporal data collection techniques (eg, drawing round-shaped timetables, which are hourly schedules represented in a circular format). The data collection process involves both non-human research (field observations) and human subject research. We are conducting human subject research in the order of individual in-depth interviews using cognitive maps and round-shaped timetables, followed by mobile GPS tracking (figure 3). Data collection has been ongoing since September 2022.

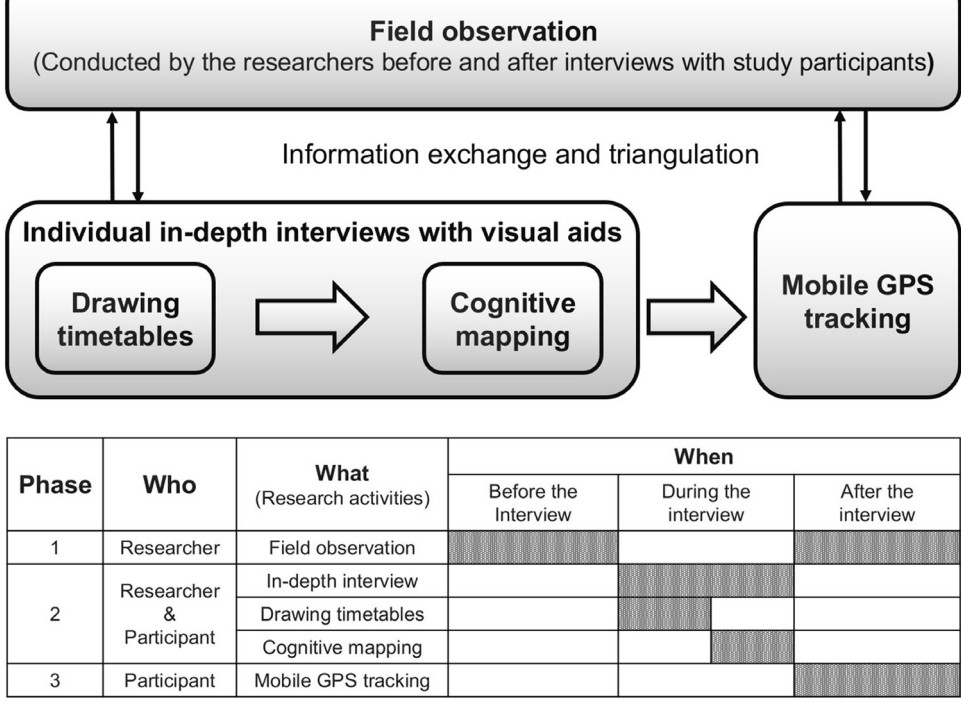

**Figure 3** Data collection process.

## Descriptive data collection

Individual in-depth interviews are an effective and proactive qualitative research method that can be used to understand the lives and experiences of research participants in their language and perspective and to discover their inner world.[28] Different types of individual in-depth interviews can be conducted depending on the means, number of research participants, interview format and openness to interview questions. The most well-known types include open, semistructured and structured interviews. For this ongoing study, we are conducting semistructured interviews with young single-person households experiencing housing poverty. The interviews are being conducted face-to-face and through the online videoconferencing platform Zoom, depending on participants' preferences. The researchers first explain the study and then obtain consent from individuals regarding their willingness to participate. Additionally, they confirm their consent to record and use their interviews, which will be transcribed and used only for the purpose of analysis. Participants are informed that they can stop the interview at any time if they wish, and the informed consent form makes it clear that participation is entirely voluntary.

After obtaining consent, we conduct a survey to collect demographic information, including sex, age, occupation, length of residence, housing characteristics and health status. Following the survey, we conduct one-on-one in-depth interviews using semistructured questions to gain insights into the perceived attributes of neighbourhood environments related to healthy lifestyles (online supplemental material 1). We record and transcribe the interviews into electronic documents with an average turnaround time of 5 days. The expected average interview time per participant is approximately 1 hour and 30 min.

## Spatial data collection

Field observation is a spatial research method in which a researcher observes phenomena related to a research topic while moving around a research site.[29] Currently, we are following a procedure for conducting field observations. We use Google Maps and Street View to examine the neighbourhood environment and plan observation paths, schedules and boundaries. To investigate the physical characteristics of the neighbourhood environment and the contents that change depending on the observation time, we use The Healthy Community Checklist.[30] We are manually recording the collected information on field observation sheets (online supplemental material 2), which will later be digitised for future reference and organised by date and neighbourhood.

Techniques to visually represent an individual or group's mental model of the physical environment are known as cognitive mapping,[31] which involves asking participants to either map from their memories or annotate existing maps with information that reflects their perceptions, knowledge and experiences of the environment. In this study, the researchers conduct activities during one-on-one interviews with the participants to create cognitive maps. Participants are asked to mark the map with reference to places and experiences related to healthy living. On completion of the cognitive map, the researchers ask the participants to explain the main activities in the marked places, their reason for selecting them and any difficulties they encountered while using them. Finally, participants are asked to indicate the environmental characteristics of their neighbourhood (within what they perceive to be its boundaries) that impact their health. We expect cognitive maps to provide insights into how people navigate and use their environments, perceive and prioritise different features and landmarks, and conceptualise their spatial relationships.

Mobile GPS can easily track the spatial movement of participants' daily activity radius in real time.[32] In this study, the participants wear the Garmin Forerunner device and record their daily lives during the survey period. The mobile GPS devices help confirm the range of participants' walking activity radii and identify their movement paths and the locations where they stayed in the neighbourhood. The participants' data, including distance travelled, duration of movement, movement path and place of residence are automatically synchronised with Google Maps and stored daily. Based on previous research evidence, the research team conducts the tracking once for a period of 3 days, including 2 weekdays and 1 day on the weekend.[33]

## Temporal data collection

During the one-on-one in-depth interviews, we use a technique that creates a round-shaped timetable to understand participants' 24-hour daily routines, which are separated into weekdays and weekends. We divide a 24-hour day into equal segments and assign activities or events to each segment based on the start and end times. Each segment is labelled and represented by a circle, similar to a clock face (online supplemental material 3). We ask participants to complete a daily schedule based on weekday and weekend routines. Once completed, the researchers review participants' schedules and ask for additional explanations of their daily neighbourhood activities. Finally, we ask participants to describe activities related to a healthy lifestyle and explain why, when and how often they engaged in them. We expect the round timetable to outline how individuals allocate their time, the duration and frequency of different activities, and the temporal patterns of their behaviour.

## Data analysis and integration

We will conduct six stages of thematic analysis[34] for the in-depth interview data. In the first stage, we will develop a general framework for participants' perceptions of the neighbourhood environment related to a healthy lifestyle by repeatedly reading the interview data. In the second stage, we will conduct an open coding process to identify all meaning units related to the perception of the neighbourhood environment, which affects participants' healthy lifestyles, such as meaning, attributes and roles. In the third stage, we will compare and contrast the

relationships between the generated codes to identify categories. In the fourth stage, we will review the tentatively derived themes to determine whether they match the research purpose and reflect the data content. In the fifth stage, we will refine the derived themes into precise definitions through a recursive process of returning data for comparative analysis. Based on the final analysis results, we will provide a detailed description and present appropriate explanations and discussions on the research purpose.

We plan to perform content analysis using spatiotemporal data. Content analysis offers a systematic and objective approach to draw accurate inferences from verbal, visual or written data, thus enabling the description and quantification of specific phenomena.[35] We will extract information, including time usage patterns (amount of time and time of day), activity types and activity range (within/outside the neighbourhood), from the timetables created by individual study participants and analyse this information through overlapping analysis. We will convert the cognitive map data provided by participants and the field observation data surveyed by the researchers into a digital map using the Quantum GIS programme in order to quantitatively analyse spatial characteristics (location, density, radius, distance and distribution). We will analyse the mobile log data collected through mobile GPS tracking for neighbourhood mobility patterns using the statistical software R and Google Maps.

During the integrated analysis phase, we plan to combine the qualitative results with living space and daily life pattern results using Atlas.ti qualitative software and a hybrid relational database. The hybrid relational database will allow us to create a descriptive data database under the attribute table of spatiotemporal data, which can integrate various types of qualitative data, such as images, texts and videos, into spatial data.[36] Atlas.ti can process text data and non-text multimedia data and be linked to GIS and codes.[37]

### Researcher characteristics and reflexivity
The research team consists of individuals aged 30 and above who are majoring in public health sciences at the university in the study area. This composition enhances the team's understanding of the neighbourhoods and facilitates participant recruitment. However, it is worth noting that none of the researchers belong to the demographic of young adults living in housing poverty, thus potentially introducing bias or leading to an overestimation in reading and interpreting the context of these young adults' lives.

### Trustworthiness
We will apply Lincoln and Guba's (1985) criteria[38] for rigour in qualitative research (credibility, transferability, dependability and confirmability) to enhance the trustworthiness of this study. To enhance the credibility of our research, we plan to cross-verify the perspectives, experiences and behaviours of individuals collected from various sources using multiple qualitative methods. To improve transferability, we will provide a detailed description of the phenomenon, enabling a comparison with background data to establish the study's context. We will offer an in-depth methodological description of the study, which promotes replicability and enhances dependability. Additionally, the study will be audited by external experts with many years of experience in qualitative research to enhance confirmability.

### Reporting of findings
The findings of this qualitative study will be reported in accordance with the Standards for Reporting Qualitative Research (SRQR) statement.[39] The application of SRQR is not confined to the writing phase alone; these standards can aid researchers in planning qualitative studies and in meticulously documenting processes and decisions made throughout the study (online supplemental material 4).

### Patient and public involvement
There is no patient and public involvement in the design, conduct, reporting or dissemination plans of this study. However, we obtain advice from key informants in the study area, including public service providers and long-term residents, to ensure pathways for participant recruitment and to enhance the research team's understanding of the local context.

## ETHICS AND DISSEMINATION
The study was approved by the Institutional Review Board of Seoul National University on 18 May 2021 (IRB number 2106/002-018). Before participating in the study, participants are asked to provide written informed consent to ensure complete anonymity and confidentiality. The informed consent form explains in detail the purpose of the recording, possible risks and discomfort, potential benefits, confidentiality of information, and withdrawal procedures. Participants' anonymity is protected, and no identifiable characteristics are specified in the recorded conversations.

We will disseminate the findings to the academic society by presenting them at international conferences on urban health and publishing them in academic journals. In addition, we plan to conduct an online seminar/webinar to share and discuss our research findings with policy-makers, researchers, community organisations and health workers who plan and implement policies and programmes for young single-person households experiencing housing poverty.

**Contributors** All authors conceived the study. All authors contributed to the development of the study design and final protocols for sample selection and data collection. DHK and JL contributed to writing the manuscript. SY provided critical editing and approved the final version of the paper.

**Funding** This work was supported by a National Research Foundation (NRF) grant funded by the Korean government (Ministry of Science, ICT, and Future Planning; NRF- 2020R1A2C2012463).

**Competing interests** None declared.

**Patient and public involvement** Patients and/or the public were not involved in the design, or conduct, or reporting, or dissemination plans of this research.

**Patient consent for publication** Not applicable.

**Provenance and peer review** Not commissioned; externally peer reviewed.

**ORCID iDs**
Dong Ha Kim http://orcid.org/0000-0001-6767-2969
Jihyun Lee http://orcid.org/0000-0001-6425-9351
Seunghyun Yoo http://orcid.org/0000-0002-9273-1761

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
