## [Reviewer comments · BMJ Open]

ARTICLE DETAILS

TITLE (PROVISIONAL)	Neighborhood environments for a healthy lifestyle among young single-person households experiencing housing poverty in Seoul, South Korea: A spatiotemporal qualitative study protocol
AUTHORS	Kim, Dong Ha; Lee, Jihyun; Yoo, Seunghyun

VERSION 1 – REVIEW

REVIEWER	Kristiansen, Maria University of Copenhagen, Department of Public Health & Center for Healthy Aging
REVIEW RETURNED	24-Aug-2023

GENERAL COMMENTS	Dear Authors, Thank you for this interesting protocol outlining a novel and important study in the field of neighborhood environments, health and poverty. The text is clear, coherent and covers all major aspects required for a protocol paper. However, the following minor issue could be clarified in order to enhance the readability and contribution of the paper: "Strengths and limitations of this study", page 2, lines 13-31: methodological weaknesses e.g. related to quality of data, complexities of combining different sources of data, or inclusion/engagement of the target population, could substitute the statement that relates more to skills in the research team ("This study requires a high level of technical expertise to collect, manage, and analyze data, which can be a barrier for researchers unskilled in GIS technology and qualitative research methods.")
---

REVIEWER	Sealey, Clive University of Worcester, Allied Health and Community
REVIEW RETURNED	20-Sep-2023

GENERAL COMMENTS	The paper provides a research protocol for a prospective research project. There is clear analysis provided to make this a relevant topic to study. The paper also provides a detailed outline of the research process. The paper is well written and easy to understand. There are a number of improvements that could be made to the paper: 1. In the Background section, when talking about young person households in Korea, the relevant comparison should have been with the Korean population as a whole, not the rest of the world, as
--

	this would enable the circumstances they experience to be more specifically contextualised. 2. Again in the Background, the analysis focusses on detailing negative aspects of the neighbourhood environment. As the focus of the paper is on health lifestyle, it would perhaps be more relevant to outline what the elements of a positive aspects of neighbourhood environment are, as this would enable a comparison to be drawn with participants. 3. The protocol details many stages for the proposed research. It is not clear, however, the order or timing of these stages. It would be good for the authors to provide a matrix to the order and timings of the stages of data collection 4. The protocol states that is is a qualitative study protocol. However, there are various quantitative stages of data collection. Does this not lean the protocol towards mixed methods? 5. In the limitations, the authors note that the study is time consuming and labour intensive, which is correct. However, they also need to note the expense associated with the study, as per the allocation of all participants with a Garmin Forerunner device. 6. The setting chosen for the study would seem to be one that is particularly disadvantaged, even in the Korean context. This needs to be acknowledged in the limitations of the study. 7. The study appears to be using stratified sampling methods, but this is not stated as such. This needs to be made clear, as well as the stratification process used. 8. The Braun and Clarke reference used is a very old one, there have been more recent updates to their thematic analysis method which could be used.
--	---

VERSION 1 – AUTHOR RESPONSE

Reviewer Comments, Author Responses and Manuscript Changes: Reviewer 1

COMMENTS TO THE AUTHOR

Thank you for this interesting protocol outlining a novel and important study in the field of neighborhood environments, health and poverty. The text is clear, coherent and covers all major aspects required for a protocol paper. However, the following minor issue could be clarified in order to enhance the readability and contribution of the paper:

STRENGTHS AND LIMITATIONS OF THIS STUDY

"Strengths and limitations of this study", page 2, lines 13-31: methodological weaknesses e.g. related to quality of data, complexities of combining different sources of data, or inclusion/engagement of the target population, could substitute the statement that relates more to skills in the research team ("This study requires a high level of technical expertise to collect, manage, and analyze data, which can be a barrier for researchers unskilled in GIS technology and qualitative research methods.")

➤ Response: Thank you for taking time to review our manuscript and giving us such insightful comments.

We found your comments immensely helpful and have revised accordingly. Detailed corrections are below point by point.

➤ Changes:

(1) STRENGTHS AND LIMITATIONS OF THIS STUDY (p2, lines 36-37)

- This study requires a high level of competence and expertise in managing data quality, addressing the complexities of integrating diverse data sources, and ensuring the active participation of the target population.

1

Reviewer Comments, Author Responses and Manuscript Changes: Reviewer 2

COMMENTS TO THE AUTHOR

The paper provides a research protocol for a prospective research project. There is clear analysis provided to make this a relevant topic to study. The paper also provides a detailed outline of the research process. The paper is well written and easy to understand. There are a number of improvements that could be made to the paper:

BACKGROUND

1. In the Background section, when talking about young person households in Korea, the relevant comparison should have been with the Korean population as a whole, not the rest of the world, as this would enable the circumstances they experience to be more specifically contextualised.

- Response: Thank you for your insightful feedback. We acknowledge the importance of providing a more context-specific comparison for young person households in Korea. In the Background section, we will revise the relevant comparison to focus on the Korean population as a whole rather than a broader comparison with the rest of the world. This adjustment will help to more accurately contextualize the circumstances experienced by young person households in the Korean context.

- Changes:

- (1) BACKGROUND (p3, lines 50-62)

Compared to the other demographic groups in Korea, young single-person households face a higher risk of health and housing problems. According to the Korean National Health and Nutrition

Examination Survey conducted in 2020, 63.2% of young single-person households aged 19–29 were either obese or overweight; 27.7% showed symptoms of depression, and 29.3% were smokers. These figures were statistically significantly higher than multi-person households (5). In 2021, the Korea Research Institute for Human Settlements (6) reported that 60% of young single-person households spent more than 30% of their monthly income on housing rent, a figure more than five times higher than that of young couple households and young households living with parents.

Additionally, according to the 2020 Korea Housing Survey, 9.6% of single-person households living in houses with less than 14 m² of residential space experienced difficulties in soundproofing, ventilation, heating, and kitchen facilities. These housing conditions are imperfect compared to the international minimum housing standards that consider detailed facilities and layouts such as kitchens, bathrooms, bathing facilities, privacy protection, appropriate hot water supply temperatures, and 25 m² per person. This suggests that young single-person households in Korea may not have sufficient resources within their homes to manage and protect their health adequately

2. Again in the Background, the analysis focusses on detailing negative aspects of the neighbourhood environment. As the focus of the paper is on health lifestyle, it would perhaps be more relevant to outline what the elements of a positive aspects of neighbourhood environment are, as this would enable a comparison to be drawn with participants.

- Response: Thank you for your valuable feedback. We recognize the need to achieve a more balanced perspective in the analysis of this study by including the positive aspects of the neighborhood environment. To address this, we will enhance the focus of our analysis in the Background section by incorporating details about the positive elements of the neighborhood environment.

- Changes:

(1) BACKGROUND (p3, lines 62-65)

Given the challenges faced by these households, namely, insufficient home resources to adequately address health needs, highlighting the health-promoting factors of the neighborhood environment could be a pivotal strategy for improving the quality of life of those living in single-person households.

(2) BACKGROUND (p4, lines 93-94)

3) What positive attributes and characteristics of neighborhood environments are necessary to support healthy lifestyles in young single-person households experiencing housing poverty?

METHODS AND ANALYSIS

3. The protocol details many stages for the proposed research. It is not clear, however, the order or timing of these stages. It would be good for the authors to provide a matrix to the order and timings of the stages of data collection.

➤ Response: Thank you for your valuable feedback on incorporating a time-ordered matrix to address concerns about the clarity of the protocol steps. Your insights have significantly contributed to enhancing the visual representation of the research plan. We appreciate your constructive comments, and the revised Figure 3 now provides a clearer understanding of the sequence and timing of the various stages of the proposed study.

➤ Changes:

(1) METHODS AND ANALYSIS (p6, line 158)

[Figure 3]

4. The protocol states that is a qualitative study protocol. However, there are various quantitative stages of data collection. Does this not lean the protocol towards mixed methods?

➤ Response: Taking note of your observation, we recognize that our research protocol might give the impression of a mixed-methods study, considering the simultaneous collection of descriptive and quantitative information. However, what we want to emphasize is that the research design itself is not based on mixed methods but rather on a qualitative GIS approach that seamlessly integrates spatial-temporal information with descriptive data. Despite incorporating quantitative aspects in the data collection stages, our research fundamentally relies on qualitative inquiry, with a dedicated focus on integrating spatial and narrative information in the research design. To avoid any potential

misunderstanding, we have updated the methodology section to explicitly state that our research does not fall under the category of mixed methods.

➤ Changes:

(1) METHODS AND ANALYSIS (p5, lines 109-115)

Figure 1 shows a schematic diagram of the extended QGIS used in this study. This diagram demonstrates how qualitative, spatial, and temporal data can be combined through simultaneous processing to produce a rich contextual understanding of people's lived experiences in their neighborhoods. This distinctive aspect sets our study's strategy apart from a mixed methods approach (20) as it has a unique sequence and arrangement for collecting both qualitative and quantitative data. Additionally, we take a more qualitative-oriented approach compared to traditional QGIS by interpreting GIS information based on patterns rather than concentrating solely on size, frequency, and density.

(2) REFERENCE (p13, lines 328-329)

20 Creswell, JW. A concise introduction to mixed methods research. Thousand Oaks, CA: Sage publications 2014:34-50.

5. The study appears to be using stratified sampling methods, but this is not stated as such. This needs to be made clear, as well as the stratification process used.

➤ Response: Thank you for your keen observation. We appreciate your feedback regarding the perceived use of stratified sampling methods in our study. You are correct in noting that this aspect should be explicitly stated for clarity. We will update our methodology section to clearly mention the use of stratified sampling.

➤ Changes:

(1) METHODS AND ANALYSIS (p6, lines 132-134)

We are currently in the planning phase of conducting stratified purposive sampling (22) for single-person households (aged 19–39) experiencing housing poverty within the study areas.

(2) REFERENCE (p13, line 332)

22 Patton MQ. Qualitative research and evaluation methods (3rd ed.). Thousand Oaks, CA: Sage 2002:240.

6. The Braun and Clarke reference used is a very old one, there have been more recent updates to their thematic analysis method which could be used.

➤ Response: Thank you for your input. In accordance with your feedback, we have updated the reference to Braun and Clarke to a more recent reference, taking into consideration the recent updates to their thematic analysis method. This change is expected to ensure that the discussed content is more contemporary and accurately reflected.

➤ Changes:

(1) REFERENCE (p14, line 351)

32 Braun V, Clarke V. Conceptual and design thinking for thematic analysis. Qual Psychol 2022;9(1): 3-6.

STRENGTHS AND LIMITATIONS OF THIS STUDY

7. In the limitations, the authors note that the study is time consuming and labour intensive, which is correct. However, they also need to note the expense associated with the study, as per the allocation of all participants with a Garmin Forerunner device.

➤ Response: Thank you for bringing attention to the limitations of our study. We appreciate your observation regarding the time-consuming and labor-intensive nature of the research, which is duly noted. Your suggestion to include a mention of the associated expenses, particularly in terms of allocating Garmin Forerunner devices to all participants, is valid. We will incorporate this aspect into the limitations section to provide a more comprehensive understanding of the study's constraints.

➤ Changes:

(1) STRENGTHS AND LIMITATIONS OF THIS STUDY (p2, lines 34-35)

➤ This study may be time-consuming, labor-intensive, and costly due to the need to collect and analyze data from multiple sources and devices.

8. The setting chosen for the study would seem to be one that is particularly disadvantaged, even in the Korean context. This needs to be acknowledged in the limitations of the study.

➤ Response: Thank you for highlighting this important consideration. We acknowledge your observation regarding the chosen setting for our study and appreciate your suggestion to explicitly acknowledge this in the limitations section. In line with your feedback, we will emphasize the need for results and interpretations based on the Korean context. This acknowledgment will be incorporated into the limitations section, providing a more transparent understanding of the study's contextual constraints.

➤ Changes:

(1) STRENGTHS AND LIMITATIONS OF THIS STUDY (p2, lines 38-39)

➤ The interpretation of the study results should consider the characteristics and context of both the study area and participants, particularly in relation to housing poverty.

1

VERSION 2 – REVIEW

REVIEWER	Sealey, Clive University of Worcester, Allied Health and Community
REVIEW RETURNED	28-Nov-2023

GENERAL COMMENTS	1. The authors have made a revision to the manuscript in relation healthy lifestyle, but this needs to be elaborated on more. Specifically, the authors could include a paragraph on what a healthy lifestyle looks like. This would enable the reader to understand what the research is aiming to achieve. At the moment, the focus is on the negative aspects of individuals, not on the healthy lifestyles that the research is trying to achieve. 2. A minor change to the Figure 3 would be to number the stages in terms of the order that they occur, as this would make the process clearer.
---

VERSION 2 – AUTHOR RESPONSE

Reviewer Comments, Author Responses and Manuscript Changes: Reviewer 2

COMMENTS TO THE AUTHOR

BACKGROUND

1. The authors have made a revision to the manuscript in relation healthy lifestyle, but this needs to be elaborated on more. Specifically, the authors could include a paragraph on what a healthy lifestyle looks like. This would enable the reader to understand what the research is aiming to achieve. At the moment, the focus is on the negative aspects of individuals, not on the healthy lifestyles that the research is trying to achieve.

➤ Response: Thank you for your valuable feedback. In response to your suggestion, we have incorporated an additional paragraph in the manuscript that provides a comprehensive overview of what a healthy lifestyle entails. This inclusion aims to shift the focus from the negative aspects of individuals to a more balanced perspective, allowing readers to better understand the positive outcomes the research seeks to achieve. If you have any further recommendations or specific points you would like us to address, please feel free to provide additional guidance. We appreciate your thoughtful input and are committed to enhancing the clarity and relevance of our research.

➤ Changes:

(1) BACKGROUND (p3, lines 66-74)

Recognizing the unique challenges faced by young, single-person households underscores the urgency of prioritizing and cultivating a healthy lifestyle. A healthy lifestyle, intricately intertwined with the surrounding environment, is a holistic concept that transcends physical health, encompassing mental health, nutrition, and overall quality of life (9). The combination of factors such as regular exercise, a balanced diet, mental well-being, and positive social interactions not only contributes to individual well-being but also interfaces with the neighborhood environment (9). This symbiotic relationship operates synergistically within the holistic framework of a healthy lifestyle. Recognizing the interconnectedness of these elements is critical, as together they form the foundation for wellness and fortification against potential health risks, with the neighborhood environment playing a pivotal role in shaping and supporting these health-promoting behaviors (10).

METHODS AND ANALYSIS

2. A minor change to the Figure 3 would be to number the stages in terms of the order that they occur, as this would make the process clearer.

➤ Response: Thank you for your suggestion. We have implemented a minor revision to Figure 3 by numbering the stages in accordance with the order of occurrence. This adjustment aims to enhance the clarity of the process depicted in the figure. If you have any further recommendations or specific details you would like us to address, please feel free to provide additional guidance. We appreciate your attention to detail and constructive feedback in refining the visual presentation of our work.

➤ Changes:

(1) METHODS AND ANALYSIS (p7, line 167)

[Figure 3]

1

Phase	Who	What (Research activities)	When		
			Before the Interview	During the interview	After the interview
1	Researcher	Field observation			
2	Researcher & Participant	In-depth interview			
		Drawing timetables			
		Cognitive mapping			
3	Participant	Mobile GPS tracking